# The Unexpected Role of the Endothelial Nitric Oxide Synthase at the Neurovascular Unit: Beyond the Regulation of Cerebral Blood Flow

**DOI:** 10.3390/ijms25169071

**Published:** 2024-08-21

**Authors:** Giorgia Scarpellino, Valentina Brunetti, Roberto Berra-Romani, Giovambattista De Sarro, Germano Guerra, Teresa Soda, Francesco Moccia

**Affiliations:** 1Laboratory of General Physiology, Department of Biology and Biotechnology “L. Spallanzani”, University of Pavia, 27100 Pavia, Italy; giorgia.scarpellino@unipv.it (G.S.); valentina.brunetti01@universitadipavia.it (V.B.); 2Department of Biomedicine, School of Medicine, Benemérita Universidad Autónoma de Puebla, Puebla 72410, Mexico; rberra001@hotmail.com; 3Department of Health Sciences, University of Magna Graecia, 88100 Catanzaro, Italy; desarro@unicz.it (G.D.S.); teresa.soda@unicz.it (T.S.); 4Department of Medicine and Health Sciences “V. Tiberio”, University of Molise, 86100 Campobasso, Italy; germano.guerra@unimol.it

**Keywords:** endothelial nitric oxide synthase, nitric oxide, cerebrovascular endothelial cells, neurovascular unit, neurovascular coupling, long-term potentiation, vascular-to-neuronal communication

## Abstract

Nitric oxide (NO) is a highly versatile gasotransmitter that has first been shown to regulate cardiovascular function and then to exert tight control over a much broader range of processes, including neurotransmitter release, neuronal excitability, and synaptic plasticity. Endothelial NO synthase (eNOS) is usually far from the mind of synaptic neurophysiologists, who have focused most of their attention on neuronal NO synthase (nNOS) as the primary source of NO at the neurovascular unit (NVU). Nevertheless, the available evidence suggests that eNOS could also contribute to generating the burst of NO that, serving as volume intercellular messenger, is produced in response to neuronal activity in the brain parenchyma. Herein, we review the role of eNOS in both the regulation of cerebral blood flow and of synaptic plasticity and discuss the mechanisms by which cerebrovascular endothelial cells may transduce synaptic inputs into a NO signal. We further suggest that eNOS could play a critical role in vascular-to-neuronal communication by integrating signals converging onto cerebrovascular endothelial cells from both the streaming blood and active neurons.

## 1. Introduction

The assignment of the 1998 Nobel Prize in Physiology or Medicine to Robert F. Furchgott [1], Louis J. Ignarro [2], and Ferid Murad [3] “for their discoveries concerning nitric oxide (NO) as a signaling molecule in the cardiovascular system” sparked an avalanche of studies aiming at investigating the pleiotropic role of this versatile gasotransmitter [4]. It was rapidly established that NO is also critical in the central nervous system (CNS) by regulating multiple brain functions, including cerebral blood flow (CBF), blood-brain barrier (BBB) permeability, synaptic plasticity, and memory formation [5,6,7,8]. Endogenous NO is generated by the catalytic activity of three isoforms of the NO synthase (NOS) [6,7,8,9]: neuronal NOS (nNOS), inducible NOS (iNOS), and endothelial NOS (eNOS), which are, respectively, encoded by the *NOS1*, *NOS2*, and *NOS3* genes. Early studies suggested that nNOS was the main source of NO at the neurovascular unit (NVU), which represents the minimal functional unit in the brain and is composed of endothelial cells, mural cells, astrocyte end-feet, and neurons [10]. However, recent investigations based upon previous work in blood vessels unveiled the critical contribution of eNOS to NO signaling at the NVU [11]: eNOS can be recruited following synaptic activity to cause a local increase in CBF [12,13], thereby regulating neurovascular coupling (NVC), i.e., the mechanism by which an increase in neuronal activity is translated into an increase in local CBF [10]. Furthermore, eNOS-derived NO may modulate synaptic activity and long-term potentiation (LTP) by establishing an unexpected line of communication between microvascular endothelial cells and neurons [14,15,16,17], which falls within the broader concept of vascular-to-neuronal communication [18,19,20]. Therefore, endothelium-derived NO does not only serve as a volume intracellular messenger that supports ongoing neuronal function through NVC, but it can also participate in learning and memory formation [16,17,21]. 

Herein, we briefly summarize the anatomical organization of the brain microcirculation and the cellular organization of the NVU. Then, we survey the cellular localization of eNOS at the NVU, the molecular mechanisms of eNOS activation, and its downstream signaling pathways. Finally, we describe the mounting evidence that eNOS can be recruited by neuronal activity to regulate CBF and LTP. The unexpected implication of these novel pieces of information is that cerebrovascular endothelial cells, which are the main site of eNOS expression at the NVU [16,22,23,24], may fine-tune neuronal activity by both triggering NVC and directly regulating synaptic plasticity.

## 2. Cerebral Circulation and the NVU

An anastomotic ring at the base of the brain, known as the Willis circle, gives rise to the three main pairs of supplying arteries of the skull, known as the anterior, middle, and posterior cerebral arteries. The circle of Willis branches out into the main intracerebral arteries that ramify into progressively smaller arteries and arterioles that supply a large territory of the cerebral cortex en route to the cortical surface. Herein, a heavily interconnected network of pial arteries and arterioles runs on the surface of the brain within the subarachnoid space, thereby originating the penetrating arterioles, which dive into the brain parenchyma. The penetrating arterioles are lined by a single endothelial cell layer, which is covered by a thin basement membrane, enwrapped by 1–3 layers of vascular smooth muscle cells (VSMCs), and ensheathed by the pia mater. The outer limit of the perivascular space (or Virchow–Robin space) that surrounds the penetrating arterioles branching off the subarachnoid space and is filled with the cerebrospinal fluid (CSF) is formed by the astrocytic end-feet of the glia limitans. The vascular basement membrane and the glia limitans fuse together, thus obliterating the perivascular space, as the penetrating arterioles penetrate downward through the cerebral parenchyma and become intraparenchymal arterioles. At this level, cerebral microvessels present one single or discontinuous layer of VSMCs and are encased by the astrocytic end-feet (Figure 1) [5,10]. Intraparenchymal arterioles ramify into a heavily intercommunicating network of capillaries, which come in close contact with the neurons (<15 μm) and are therefore in the most suitable position to detect neuronal activity [25,26]. The capillary wall of the BBB consists of an endothelial monolayer that lacks VMSCs but is partially surrounded by contractile pericytes, which are also embedded into the vascular membrane basement and cover ≈30% of the capillary surface [27]. Astrocytic end-feet may cover the remaining 70% of the brain capillary surface and regulate the induction and maintenance of BBB properties (Figure 1) [10,27]. 

Neurons are obviously key cellular constituents of the NVU. The “extrinsic innervation” of the pial surface vasculature is brought about by sympathetic nerves from the superior cervical ganglion, parasympathetic nerves from the sphenopalatine and otic ganglia, and the trigeminal ganglion. The “extrinsic innervation” regulates the contractile tone of pial VSMCs by releasing multiple neurotransmitters and neuromodulators, including norepinephrine, acetylcholine, vasoactive intestinal peptide (VIP), neuropeptide Y, calcitonin gene-related peptide (CGRP), and NO itself, and is primarily responsible for shifting the upper limit of the myogenic response, also known as autoregulation, toward higher pressures [28,29]. Penetrating arterioles are barely innervated by extrinsic fibers, while intraparenchymal arterioles receive extensive innervation by local γ-aminobutyric acid (GABA) interneurons, often with the interposition of astrocytes. In addition, intraparenchymal arterioles receive neuronal inputs from sub-cortical neurons that are located in the basal forebrain, raphe nucleus, and basal forebrain (Figure 1) [28,29]. The primary role of this “intrinsic innervation” is to adjust the intrinsic vasomotor activity of intracerebral microvessels to neuronal activity, according to a mechanism known as NVC [28,29]. Therefore, brain capillary endothelial cells and local pyramidal neurons/interneurons can feed back onto one another, with the BBB regulating the supply of oxygen (O_2_) and nutrients to active neurons and the latter delivering signaling mediators (e.g., neurotransmitters and neuromodulators) to the vascular cells.

## 3. NOS Isoforms and NO Signaling at the NVU

The discovery of NO as a signaling messenger in the brain has taken advantage of the identification of NO as a vasorelaxing mediator in the cardiovascular system [30,31]. This observation supports the view, more broadly discussed below, that eNOS, i.e., solely responsible for NO production in blood vessels, could play a role that goes beyond the regulation of BBB permeability and NVC [11]. The endogenous synthesis of NO at the NVU is strictly controlled by the constitutive NOS isoforms (cNOSs), i.e., nNOS and eNOS, which are both activated upon an increase in intracellular Ca^2+^ concentration ([Ca^2+^]_i_) and the recruitment of the Ca^2+^/dependent calmodulin (CaM). Conversely, iNOS, which is constitutively bound to CaM and is therefore Ca^2+^-independent, is barely detectable in the CNS but can be expressed in astrocytes, microglia, and blood-derived macrophages in response to an immune challenge and will not be further addressed here [7,30,31].

### 3.1. nNOS and eNOS: Cellular Localization within the NVU and Mechanisms of Synaptic Activation 

nNOS can be activated by glutamate release during synaptic activity. The expression pattern of nNOS is restricted to discrete neuronal subtypes, such as inhibitory interneurons and a small population of excitatory neurons in the cerebral cortex, striatum, hippocampus, cerebellum, and olfactory bulb [22,30,32,33,34]. Glutamate triggers Ca^2+^ influx through N-methyl-D-aspartate (NMDA) receptors (NMDARs) [32,35], thereby leading to a robust increase in dendritic Ca^2+^ concentration that engages CaM, which in turn binds to nNOS and accelerates NO production by facilitating interdomain electron transfer (Figure 2) [7,30,31]. The nNOS protein, via the scaffolding protein postsynaptic density-95 (PSD-95), is physically coupled with the GluN2B subunit of NMDARs; this ternary complex, nNOS–PSD95-NMDAR, ensures the efficiency and fidelity of the signal transduction pathway responsible for NO release in response to synaptic activation (Figure 2) [7,30,31].

Although early studies claimed that it could also be expressed in neurons [36,37,38], there is now no doubt that eNOS is primarily located within the endothelial component of the NVU [16,22,23,24]. Moreover, eNOS is also expressed in pericytes [39], although its functional role is yet to be unraveled [40]. An increase in [Ca^2+^]_i_ is the main signal responsible for eNOS activation in endothelial cells throughout the vascular tree, including the brain microcirculation [11,41,42]. Interestingly, cerebrovascular endothelial cells also express NMDARs that can be stimulated by synaptically released glutamate to promote NO release (Figure 2) [11,43,44]. A recent study confirmed that Ca^2+^ entry through NMDARs may cause a spatially restricted sub-membrane Ca^2+^ domain that also elicits NO release in the human cerebrovascular endothelial cell line, hCMEC/D3 [45]. It should be noted that, in the other vascular districts, Ca^2+^ entry through store-operated Ca^2+^ channels (SOCs) is the primary stimulus responsible for eNOS activation in response to chemical stimulation [41,42,46]. However, an in vivo study revealed that synaptic activity stimulates G_q_-coupled protein receptors (G_q_PCRs) that lead to endothelial Ca^2+^ oscillations and NO release by activating endoplasmic reticulum (ER)-dependent Ca^2+^ release through inositol-1,4,5-trisphosphate (InsP_3_) receptors (InsP_3_Rs) and extracellular Ca^2+^ entry through Transient Receptor Potential (TRP) Vanilloid 4 (TRPV4) [47]. It has been suggested that synaptically elicited endothelial Ca^2+^ oscillations could be driven by mGluR1 and mGluR5, which are G_q_PCRs widely expressed in cerebrovascular endothelial cells [48,49]. Subsequent investigations showed that other neurotransmitters and neuromodulators, including acetylcholine, GABA, and histamine, were also able to induce oscillatory or biphasic endothelial Ca^2+^ signals that led to NO release in vitro [48,50,51,52,53,54]. Finally, an additional source of Ca^2+^ for eNOS activation at the NVU could be provided by the mechanosensitive Piezo1 channels [55], which are activated by laminar shear stress and are coupled to eNOS in multiple vascular districts [56]. Therefore, it is likely that any increase in [Ca^2+^]_i_ can recruit eNOS in cerebrovascular endothelial cells. 

Throughout the peripheral circulation, eNOS is spatially segregated in caveolae [41,42], where it is tonically inhibited by the physical association with caveolin-1 (CaV-1) and is bound to the plasma membrane due to NH_2_-terminal fatty acylation by palmitic and myristic acid [21,57]. The increase in endothelial [Ca^2+^]_i_ serves to stimulate the Ca^2+^/CaM to displace CaV-1 from the Ca^2+^-binding domain and activate eNOS [21]. However, a recent investigation showed that eNOS signaling and caveolae-mediated pathways may not overlap in the mouse brain microcirculation [12]. This finding prompts the search for alternative mechanisms of Ca^2+^/CaM-dependent eNOS activation in cerebrovascular endothelial cells. Interestingly, eNOS can also be activated upon phosphorylation by different signaling pathways, including phosphoinositide 3 kinase (PI3K)/protein kinase B (PKB/Akt), extracellular regulated protein kinases 1/2 (ERK1/2)/mitogen-activated protein kinase (MAPK), and cyclic adenosine monophosphate/protein kinase A (cAMP/PKA) [21,57]. Furthermore, NO release from endothelial cells could also be elicited by Ca^2+^/CaM-dependent protein kinase II (CaMKII)-mediated eNOS phosphorylation [58,59]. This evidence suggests that synaptic activity or chemical stimulation could activate eNOS by causing an increase in CaMKII activity, which could overcome the lack of PSD95. 

### 3.2. eNOS Signaling within the NVU 

eNOS (as well as nNOS and iNOS) is an oxidoreductase that catalyzes the oxidation of L-arginine to NO and L-citrulline, with nicotinamide adenine dinucleotide phosphate (NADPH), flavin mononucleotide (FMN), and flavin adenine dinucleotide (FAD) participating in the electron transfer chain and O_2_ serving as the final electron acceptor (Figure 2). The cofactor tetrahydrobiopterin (BH4) accelerates the rates and improves the efficiency of the electron transfer, thereby ensuring that L-arginine oxidation is coupled with NO synthesis [9,21,57]. Its physicochemical properties hint at NO as a candidate volume transmitter in the brain [30,60], being able to diffuse within a finite volume in all directions away from the site of production and integrate the activity of multiple cell types, i.e., neurons, astrocytes, endothelial cells, and mural cells, irrespective of their synaptic connectivity. Of course, most of the attention has been paid to nNOS- rather than eNOS-dependent NO signaling. For instance, the activation of NMDARs with NMDA induces a transient NO signal that spreads for ≈400 μm within a radius of 100 μm from the stimulation site [61]. These measurements have been carried out in the CA1 region of the hippocampus by means of an electrochemical sensor [61] that may detect the endogenous production of NO by both nNOS and eNOS, which is quite abundant in pyramidal neurons of the CA1 region [22]. Intriguingly, the NO signal triggered by the exogenous application of NMDA was suppressed by N_G_-nitro-l-arginine (l-NNA), which blocks both isoforms [61]. In 2005, the notion that cerebrovascular endothelial cells also express NMDARs and can respond to synaptic stimulation by producing NO was yet to come [11,43,44,45,62], and the NO signal was entirely ascribed to nNOS activation [61]. Nevertheless, we can now reinterpret those pioneering findings in light of the new information and conclude that the diffusional spread of NO from the NMDA ejection site is likely to be supported by multiple NOS sources [61], including both nNOS and eNOS. 

In agreement with this hypothesis, Garthwaite has recalled that the prototypic example of volume transmission is provided by NO signaling in the blood vessels, where NO generated in endothelial cells diffuses to the overlying VSMCs to cause vasorelaxation [30,31,63]. In addition, Garthwaite has already pinpointed the potential role of eNOS in volume transmission, especially at the capillary level, where the density of the microvascular network is such as to boost NO production induced by synaptic activity [30]. Therefore, eNOS could integrate the endothelial Ca^2+^ signals deriving from the stimulation of both abluminal NMDARs and G_q_PCRs (which receive synaptic contacts and can be activated by neurotransmitters and neuromodulators) and luminal G_q_PCRs and mechanosensitive channels (which detect blood-borne chemical and physical signals). The ensuing NO signal generated by the simultaneous activation of a cluster of endothelial cells [42,64,65,66] could then spread from its site of synthesis to the other cellular constituents of the vascular wall and to the brain parenchyma. This would remarkably extend the spatial range of action of endothelial NO, which could not only regulate VSMC/pericyte contraction and endothelial permeability but also neuronal activity, axonal excitability, and synaptic plasticity [11,17,30]. As anticipated above, the capillary district of the bran microcirculation is ideally suitable for volume transmission by NO. Herein, virtually every neuron has its own capillary, and when synaptic stimulation coordinates the activity of a population of neurons and endothelial cells that are closely packed within the same volume of brain tissue, the recruitment of multiple NO sources (i.e., eNOS and nNOS) may result in the rapid buildup of NO in between those cells, thereby also influencing the activity of more distant cells (neurons, astrocytes, and vascular cells).

### 3.3. The Regulation of CBF by eNOS 

It has long been known that NO plays a primary role in NVC, also known as functional hyperemia [10,26,32]. Glutamate is the most abundant excitatory neurotransmitter in the brain and, therefore, is the best-characterized stimulus for NVC [10]. For the reasons highlighted in Chapter 3.2, most studies focused on nNOS activation by neuronal NMDARs followed by NO-mediated vasorelaxation of the intraparenchymal arterioles [5,10,32]. NO diffuses to the overlying VSMCs to activate the stimulated soluble guanylyl cyclase (sGC) and induce the production of cyclic guanosine-3′,5′-monophosphate (cGMP) (Figure 3). cGMP, in turn, stimulates cGMP-dependent protein kinase (PKG), which causes VSMC relaxation by phosphorylating multiple targets (Figure 3) [26]. Furthermore, synaptically released NO could also relax precapillary sphincters and pericytes at first-order capillaries, thereby enhancing blood perfusion to downstream microvessels [67]. A non-classical mechanism by which NO may regulate functions other than the vascular tone, e.g., DNA repair, transcription, and cell growth, is achieved through post-translational modification of target proteins. Non-classical NO signaling includes S-nitrosylation of protein thiols, oxidative nitration, hydroxylation, S-glutathionylation, and metal nitrosylation of transition metals [68,69].

Endothelium-derived NO was claimed to be primarily involved in the hemodynamic response to the stimulation of basal forebrain acetylcholine neurons, which broadly project onto the intraparenchymal arterioles and capillaries of the cortex [21,26,28]. In line with this, the acetylcholine-induced increase in CBF is impaired by the genetic deletion of eNOS [70] and M5 muscarinic receptors [71,72], which can activate eNOS via a long-lasting increase in endothelial [Ca^2+^]_i_ [50]. Furthermore, eNOS has been involved in the hemodynamic response of the somatosensory cortex to whisker stimulation, which triggers ATP release from synaptically activated astrocytes [13]. Then, ATP binds to P2Y1 receptors on the adjacent endothelial cells, thereby promoting eNOS activation and the local increase in CBF [13] through the elevation in endothelial [Ca^2+^]_i_ [73].

The discovery that NMDARs are also expressed in cerebrovascular endothelial cells led to a reappraisal of the cellular and molecular mechanisms of NVC at glutamatergic synapses [11,74,75]. In accordance, a series of studies showed that the exogenous administration of glutamate stimulated endothelial NMDARs to promote NO-mediated arteriolar relaxation in ex vivo mouse middle cerebral arteries [43,62,76]. A neuro-glial-endothelial axis provides endothelial NMDARs with the co-agonist D-serine in the mouse brain microcirculation [43], while the molecular make-up of human NMDARs does not support a critical role for either D-serine or glycine [11,45,75]. NMDARs are located in the abluminal membrane and, therefore, are ideally positioned to detect synaptic activity and recruit eNOS [43]. In line with these observations, the hemodynamic response to sensory stimulation is impaired by the genetic deletion of both endothelial NMDARs [44] and eNOS [12]. It follows that eNOS interacts with nNOS to regulate NVC, thereby complementing each other and providing redundancy to ensure that active neurons receive a sufficient amount of nutrients and O_2_ to maintain cerebral function [77]. Furthermore, studies carried out on healthy adult subjects suggest that both eNOS and nNOS contribute to generating the NO signal driving NVC in humans [78,79,80]. Interestingly, it has been proposed that basal CBF is not reduced in eNOS knockout mice due to autoregulation and a compensatory increase in perivascular nNOS expression or signaling [81,82,83]. The pathological implications of defective eNOS signaling fall beyond the scope of the present article [17,21,81]. However, eNOS-deficient mice, which are more prone to developing spontaneous hypertension and other defects in systemic circulation [84,85], also show an increase in infarct volume and reduced arteriogenesis after stroke [86]. Interestingly, genetic screening showed that polymorphisms in the *NOS3* gene were associated with an increased risk of cerebral small vessel disease in patients, including silent brain infarction [87,88]. The physiological relevance of eNOS signaling at the NVU is, therefore, further highlighted by the pathological consequences of its impairment in both human patients and transgenic mouse models.

It should also be noted that eNOS-derived NO is not the sole vasorelaxing mediator by which cerebrovascular endothelial cells regulate CBF at the NVU. A complex ion signaling machinery, reviewed in [25,48,89,90], is set in motion by neuronal activity at the capillary level, thereby ensuring that both fast (i.e., EDH) and slow (i.e., intercellular Ca^2+^ waves) vasorelaxing signals travel back to up-stream arterioles and precapillary sphincters to steal blood from silent to active brain areas [47,91,92,93,94]. Altogether, this evidence re-evaluates the crucial role of endothelial cells in the regulation of blood flow in the brain microvasculature, although neuron-derived vasorelaxing pathways provide a backup mechanism to preserve (at least partially) NVC even during endothelial dysfunction or damage [10,32,77].

### 3.4. The Regulation of Synaptic Plasticity by eNOS 

NO has been postulated to serve as a retrograde messenger at glutamatergic synapses, being produced in response to NMDARs-mediated Ca^2+^ entry and traveling back to the presynaptic terminal to increase glutamate release [31,95]. This mechanism explains the role of NO in synaptic plasticity, including LTP [63]. The source of NO during LTP induction is commonly claimed to be nNOS [6,96], but multiple pieces of evidence suggest that eNOS also generates NO during LTP in several brain regions. O’Dell and coworkers were the first to suggest that eNOS was responsible for LTP induction in the CA1 region of nNOS-deficient mice [36]. This pioneering finding was confirmed by three different research groups exploiting the adenoviral-mediated inhibition of eNOS activity [97], eNOS-deficient mice [98], and nNOS- and eNOS-deficient mice [99]. Subsequent studies showed that tetanic stimulation also requires eNOS to induce LTP in the cortex [100] and striatum [101]. It should again be noted that the interpretation of these studies was biased by the mislocalization of eNOS, which was wrongly claimed to also be expressed in dendritic spines [36]. The recognition that the immunocytochemical staining of eNOS in pyramidal neurons was artifactual, in association with more sensitive immunocytochemistry or in situ hybridization assays, unveiled the unexpected involvement of endothelium-derived NO in LTP [16,22,23,24]. Garthwaite and coworkers were the first to revise the view that cerebrovascular endothelial cells merely serve as a conduit of nutrients and O_2_ to active neurons [15]. They found that capillary endothelial cells tonically release NO to depolarize the optic nerve by activating sGC, thereby inducing the cyclic guanosin monophosphate (cGMP)-dependent activation of hyperpolarization-activated cyclic nucleotide-gated non-selective cation channels [15]. No other NOS isoforms, i.e., nNOS and iNOS, were involved in NO-dependent axonal depolarization. Furthermore, bradykinin, which is a widely employed endothelial agonist [42], boosted NO-dependent depolarization of the optic nerve [15]. A follow-up investigation shed further light on the role played by eNOS in hippocampal LTP [14]. This study revealed that LTP requires phasic bursts of NO release by nNOS, while eNOS provides the tonic NO signal (≈1 nM) that prevents LTP from rapidly decaying [14]. These findings help reconcile the double requirement for hippocampal LTP of both nNOS and eNOS that has been described in [99]. Furthermore, a recent investigation suggested that the hippocampal LTP may be supported by the neuro-glial-endothelial axis, which is indispensable for eNOS activation [102].

The concept of heart-brain axis (HBA) predicts that cardiovascular disorders, such as heart failure, atherosclerosis, and hypertension, may lead to vascular dementia by primarily causing cerebral hypoperfusion [103,104,105]. Similarly, aging and other cardiovascular risk factors, as well as environmental pollution, may cause early endothelial injury at the NVU, thereby accelerating the harmful consequences of amyloid-β (Aβ) peptide deposition and resulting in neurodegeneration and synaptic dysfunction in Alzheimer’s disease (AD) [5]. In addition, the amyloid hypothesis formulated by Selkoe [106] and Hardy [107] predicts that the excessive accumulation of the Aβ peptide in cerebral microvessels directly causes endothelial dysfunction (amyloid angiopathy) [17,108]. Intriguingly, transgenic mouse models of AD showed that cerebrovascular endothelial dysfunction appears long before the development of AD symptoms and the impairment of cognitive decline [17,21,108]. Endothelial dysfunction could certainly exacerbate cognitive decline by impairing CBF, dismantling BBB integrity, and promoting inflammation and astrogliosis [109,110]. In light of the unexpected role of eNOS in synaptic plasticity, we propose that endothelial dysfunction could impair the LTP induction mechanisms, thereby directly contributing to cognitive decline in AD disease [111]. In agreement with this hypothesis, the LTP/long-term depression (LTD) balance is shifted toward LTD in eNOS-deficient mice [101], which is a feature of AD [112]. Furthermore, the genetic deletion of eNOS has been shown to exacerbate spatial learning deficits in mouse models of vascular dementia [113] and AD [114], and in eNOS knockout mice after ischemic stroke [115].

### 3.5. The Role of eNOS in Vascular-to-Neuronal Communication 

In 2008, Moore and Cao introduced the concept of vascular-to-neuronal communication, according to which the mechanical stimulation of cerebrovascular endothelial cells during the hemodynamic response leads to a NO signal that may fine-tune neuronal activity [18]. The mechanosensitive channel activated by mechanical deformation and responsible for eNOS activation could be Piezo1 [55,56]. This concept has been reinforced by Filosa and coworkers, who showed that an increase in flow/pressure within the intraparenchymal arterioles of cortical brain slices caused an astrocyte-dependent decrease in the firing rate of pyramidal neurons [116,117]. It is unclear whether this signaling pathway requires the interposition of endothelial cells [20]; however, it lends strong support to the modulation of neuronal activity and synaptic processes by vascular inputs. On the one hand, assessing whether and how cerebrovascular endothelial cells generate NO in response to mechanical stimulation deriving from changes in the intraluminal pressure [56] or by the frictional force imposed by the train of red blood cells transiting through the capillaries in a single file [118] is a mandatory task. On the other hand, we propose to revise the concept of vascular-to-neuronal communication in light of the documented endothelial response to neuronal activity. Endothelium-derived NO may also be generated during synaptic activity, thereby serving as a volume intercellular messenger that contributes to induced and maintained LTP and targets mural cells to cause vasorelaxation and match the increased demand for blood supply (Figure 4). In addition, the burst of NO associated with eNOS activation could also modulate astrocyte activity, as envisaged by Moore and Cao [18]. It has long been known that the G_q_PCRs, mGluR1, and mGluR5, are involved in the astrocytic-dependent regulation of NVC, which occurs through the release of epoxyeicosatrienoic acids [119,120,121]. However, it is still debated whether astrocytes express mGluRs in vivo [10]. However, a series of independent studies showed that synaptic activity elicits Ca^2+^ signals in astrocytic end-feet and arteriolar vasodilation by promoting endothelial-dependent NO production (Figure 4), which could be driven by endothelial mGluR1 and mGluR5 [49,122]. Therefore, eNOS could play a critical role in integrating neuronal and vascular inputs, thereby ensuring a bidirectional mode of communication that guarantees the effective functioning of the NVU in the healthy brain microcirculation.

## 4. Conclusions

The discovery of NO as a crucial regulator of cardiovascular function has paved the way for the identification of a highly versatile gasotransmitter with broader physiological implications [68], extending to immune responses, inflammation, osteogenesis, smooth muscle contraction, and neuronal signaling. The role of eNOS at the NVU has progressively gained momentum over the last decade, as nNOS was initially considered the primary source of NO within the brain parenchyma [16]. eNOS is unlikely to be the first thought of a neuroscientist at work to unlock the mechanisms by which NO is produced in response to neuronal activity to modulate CBF and synaptic plasticity. Nevertheless, it is now clear that the complex puzzle of NO signaling at the NVU misses crucial pieces without the addition of cerebrovascular endothelial cells and eNOS. Future work will have to assess whether endothelium-derived NO regulates CBF in brain areas other than the somatosensory cortex, e.g., in the hippocampus and cerebellum. It will also be mandatory to unravel whether eNOS supports LTP, as well as other modes of NO-dependent synaptic plasticity [96], throughout all glutamatergic synapses in the brain or at specific sites, e.g., the hippocampal CA1 region. Another indispensable step in filling the gap between hypotheses and knowledge would be to confirm whether the increased blood flow through brain capillaries during the hemodynamic response stimulates NO release. Therefore, we posit that the concept of vascular-to-neuronal communication should include endothelium-derived signals, including the NO burst, produced in response to neuronal activity upon the release of multiple neurotransmitters and neuromodulators. Finally, although pericytes have long been regarded as a target of NO signaling at the NVU, under both physiological [40] and pathological [123] conditions, emerging evidence indicates that they can also express eNOS [39]. Intriguingly, brain pericytes express a wide range of G_q_PCRs that are potentially involved in vasoconstriction, such as α1-adrenoreceptors and endothelin receptor type B [40]. Excessive eNOS signaling in brain pericytes during cerebral ischemia could play a pivotal role in triggering α-smooth muscle actin phenotype transformation [39] and pericyte contraction due to oxidative-nitrosative stress [123], thereby leading to the no-reflow phenomenon and patient death. Understanding the role of eNOS in the regulation of neurovascular interactions is predicted to shed light on alternative strategies to prevent or interfere with vascular cognitive decline by targeting the NO-sGC-cGMP signaling pathway [124].

## Figures and Tables

**Figure 1 ijms-25-09071-f001:**
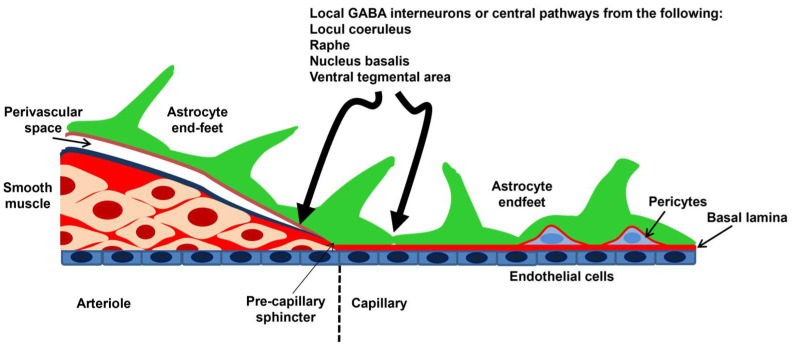
Schematic representation of the NVU. The penetrating arterioles depart from the pial arteries and are surrounded by the perivascular space, also known as Virchow–Robin space, which may house perivascular macrophages and other cell types, including mast cells and Mato cells [5,10]. Their wall includes 1–3 layers of VSMCs that determine their resistance to blood flow: an increase in the contracting state reduces blood perfusion, while a decrease in the contracting state increases blood perfusion to downstream capillaries. The outer limit of the perivascular space is lined by the astrocyte end-feet, which give rise to the glia limitans membrane. When the basal lamina fuses with the glia limitans, the perivascular space is obliterated, and the penetrating arterioles become intraparenchymal arterioles. In the capillary vessels, VSMCs are replaced by pericytes, which are contractile cells closely embedded in the vascular wall, and the outer wall is still contacted by the astrocytic end-feet. Intraparenchymal arterioles and capillaries receive intrinsic innervation from local interneurons and subcortical pathways. Modified from [26] (https://creativecommons.org/licenses/by/4.0/ (accessed on 18 June 2024)).

**Figure 2 ijms-25-09071-f002:**
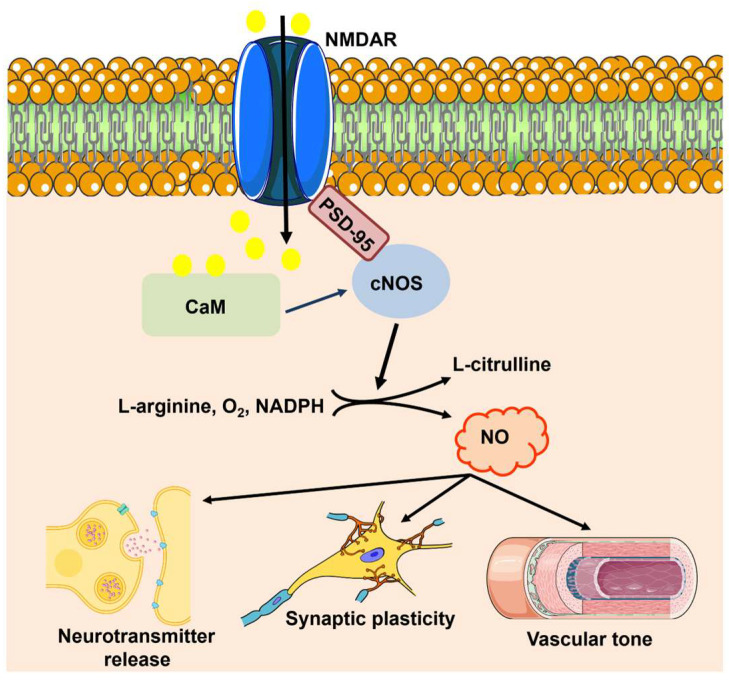
NO signaling at the NVU. Synaptic activity leads to extracellular Ca^2+^ influx through NMDARs, thereby stimulating CaM activity. Then, CaM promotes the physical association of PSD-95 to the PDZ motif in the NMDAR protein, and PSD-95 and CaM assemble into a trimeric complex with nNOS. Ca^2+^ entry through NMDARs can also activate eNOS, but it is still unclear whether this mechanism requires the involvement of PSD-95. nNOS and cNOS catalyze the 5-electron oxidation of L-arginine into L-citrulline and NO by using the cofactors NADPH and BH4. Once synthetized, NO regulates neurotransmitter release, promotes LTP, and increases local CBF by promoting VSMC and pericyte relaxation.

**Figure 3 ijms-25-09071-f003:**
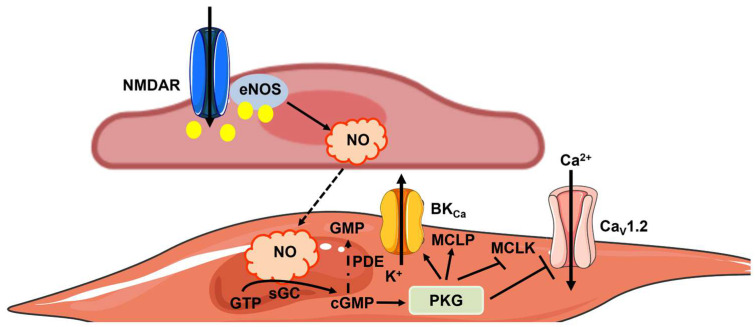
The mechanisms of NO-mediated vasorelaxation. NO can be produced by eNOS upon synaptic activation of NMDARs, although it is likely that any increase in endothelial [Ca^2+^]_i_ stimulates eNOS at the NVU [25,48]. Endothelium-derived NO may then diffuse to overlying VSMCs to induce relaxation by stimulating sGC activity, thereby resulting in GTP conversion into cGMP. In accordance, cGMP activates PKG, which in turn activates myosin light-chain phosphatase (MLCP) and inhibits myosin light-chain kinase (MLCK), thereby reducing the Ca^2+^-sensitivity of the contractile machinery. Moreover, PKG reduces VSMC excitability by activating large-conductance Ca^2+^-activated K^+^ channels (BK_Ca_) and inhibiting voltage-gated Ca_V_1.2 channels, thereby promoting VSMC hyperpolarization and preventing voltage-gated Ca^2+^ entry. The signal transduction pathway elicited by NO can be interrupted when cGMP is hydrolyzed into an inactive 5′-GMP metabolite by phosphodiesterase (PDE) activity.

**Figure 4 ijms-25-09071-f004:**
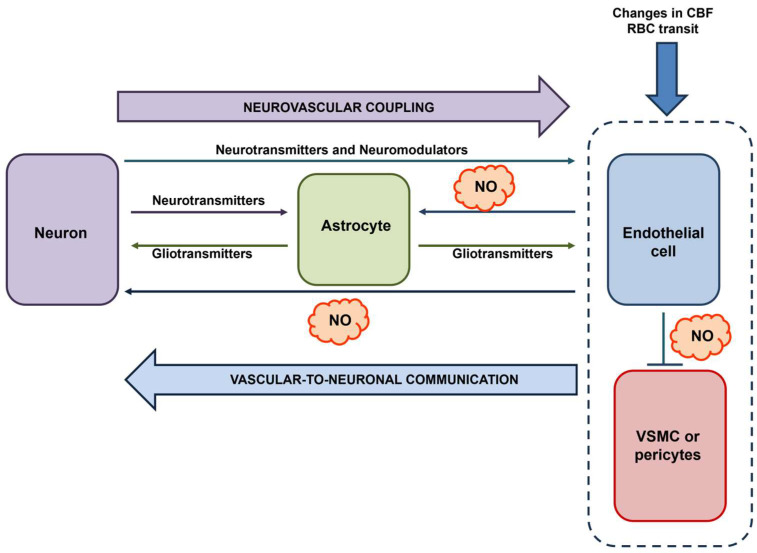
The central role of eNOS in vascular-to-neuronal communication. Neuronal activity (possibly with the intermediation of astrocyte-derived signals) and blood-born mechanical signals (e.g., local changes in CBF evoked by the same neuronal activity or single-file transit of RBCs) lead to an increase in [Ca^2+^]_i_ in cerebrovascular endothelial cells, thereby leading to eNOS activation and NO release. NO serves as a volume intercellular messenger that can target multiple cell types within the NVU: (1) neurons, thereby supporting LTP; (2) mural cells, i.e., VSMCs and pericytes, thereby promoting vasorelaxation; and (3) astrocytes, thereby eliciting intracellular Ca^2+^ signals. Inspired by [18,19].

## Data Availability

Not applicable.

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
