# Peer review of "The Unexpected Role of the Endothelial Nitric Oxide Synthase at the Neurovascular Unit: Beyond the Regulation of Cerebral Blood Flow"

_ijms, 2024, doi:10.3390/ijms25169071_

Round 1

Reviewer 1 Report

Comments and Suggestions for Authors

The manuscript: “The unexpected role of the endothelial nitric oxide synthase at the neurovascular unit: beyond the regulation of cerebral blood flow” is focusing on reviewing current state of knowledge on function of endothelial derived NO, with particular focus on signaling pathways and interactions of different components of NVU in the brain in response to NO. Figures are clear, figure legends appropriate. The paper is nicely structured, written very concisely and easy to read. Additionally, includes an overview of the past decade of knowledge with the recently published papers and I believe it is of great interest for the researchers in the field.

Minor comment, throughout the manuscript referencing to specific Figure can be decreased, for example: Lines 77-94 My personal preference is to only have one (Figure 1) at the end of paragraph. Line 105 -there is nothing on Figure 1describing GABA innervation, reference would be more appropriate here. Same, lines 166 and 210 do not need Figure 2, just references, similar for Fig 4.

Author Response

Dear Reviewer #1,

We truly thank you for your insightful comments on our manuscript entitled: “The unexpected role of the endothelial nitric oxide synthase at the neurovascular unit: beyond the regulation of cerebral blood flow” submitted for publication as Review Article in International Journal of Molecular Sciences – Special Issue “The 25th Anniversary of NO”.

We carefully addressed all your concerns, which significantly improved the quality of our manuscript.

More specifically:

Minor comment, throughout the manuscript referencing to specific Figure can be decreased, for example: Lines 77-94 My personal preference is to only have one (Figure 1) at the end of paragraph. Line 105 -there is nothing on Figure 1describing GABA innervation, reference would be more appropriate here. Same, lines 166 and 210 do not need Figure 2, just references, similar for Fig 4.

We reduced the number of the references to all the Figures throughout the manuscript, as kindly suggested by the Reviewer. Moreover, we have modified Figure 1 to include the description of local GABAergic interneurons.

We therefore hope that the manuscript will now be regarded worth of being published on this thrilling special issue of International Journal of Molecular Sciences.

Sincerely,

Francesco Moccia

Reviewer 2 Report

Comments and Suggestions for Authors

The work needs to include more mechanistic detail about the different pathway downstream NO production. For example there is not mention to the S-nitrosylation pathway that constitute a very significant mechanism related to the NO concentrations. Also there is not mention to possible role of NO produced by eNOS inastrocyte and pericytes which are part of the NVU.

Author Response

Dear Reviewer #2,

We truly thank you for your insightful comments on our manuscript entitled: “The unexpected role of the endothelial nitric oxide synthase at the neurovascular unit: beyond the regulation of cerebral blood flow” submitted for publication as Review Article in the International Journal of Molecular Sciences – Special Issue “The 25th Anniversary of NO”.

We carefully addressed all your concerns, which significantly improved the quality of our manuscript.

More specifically:

For example there is not mention to the S-nitrosylation pathway that constitute a very significant mechanism related to the NO concentrations. Also there is not mention to possible role of NO produced by eNOS inastrocyte and pericytes which are part of the NVU.

We thank the Reviewer for these observations.

We have briefly discussed the non-canonical mechanisms of NO signaling, including S-nitrosylation, in the revised manuscript, lines 272-276. We did not discuss the cytotoxic effects of excessive NO production as this information was far beyond the scope of the present manuscript, which aimed at highlighting the physiological (rather than pathological) roles of eNOS at the neurovascular unit.

Conversely, there is widespread agreement that eNOS is not expressed in astrocytes, which rather express iNOS. This information has been provided in the original version of the manuscript (lines 140-143, now highlighted in red). Similarly, there is not much evidence in support of eNOS expression in brain pericytes, which are regarded more as a target than as a source of NO signaling at the neurovascular unit (PMID: 36763972). However, in order to follow the Reviewer’s suggestion, we have now briefly addressed these important issues in lines 170-171 and lines 452-460.

We therefore hope that the manuscript will now be regarded worth of being published on this thrilling special issue of International Journal of Molecular Sciences.

Sincerely,

Francesco Moccia